# In situ strategy for biomedical target localization via nanogold nucleation and secondary growth

Akira Sawaguchi [1 ✉], Takeshi Kamimura[2], Nobuyasu Takahashi[1], Atsushi Yamashita[3], Yujiro Asada[3], Hiroyuki Imazato[4], Fumiyo Aoyama[1], Akiko Wakui[2], Takeshi Sato[2], Narantsog Choijookhuu[5] & Yoshitaka Hishikawa[5]

Immunocytochemistry visualizes the exact spatial location of target molecules. The most common strategy for ultrastructural immunocytochemistry is the conjugation of nanogold particles to antibodies as probes. However, conventional nanogold labelling requires time-consuming nanogold probe preparation and ultrathin sectioning of cell/tissue samples. Here, we introduce an in situ strategy involving nanogold nucleation in immunoenzymatic products on universal paraffin/cryostat sections and provide unique insight into nanogold development under hot-humid air conditions. Nanogold particles were specifically localized on kidney podocytes to target synaptopodin. Transmission electron microscopy revealed secondary growth and self-assembly that could be experimentally controlled by bovine serum albumin stabilization and phosphate-buffered saline acceleration. Valuable retrospective nanogold labelling for gastric $H^+/K^+$-ATPase was achieved on vintage immunoenzymatic deposits after a long lapse of 15 years (i.e., 15-year-old deposits). The present in situ nanogold labelling is anticipated to fill the gap between light and electron microscopy to correlate cell/tissue structure and function.

[1] Division of Ultrastructural Cell Biology, Department of Anatomy, University of Miyazaki, Miyazaki, Japan. [2] Hitachi High-Tech Corporation, Tokyo, Japan. [3] Division of Pathophysiology, Department of Pathology, University of Miyazaki, Miyazaki, Japan. [4] Division of Orthopedic Surgery, Department of Medicine of Sensory and Motor Organs, University of Miyazaki, Miyazaki, Japan. [5] Division of Histochemical Cell Biology, Department of Anatomy, University of Miyazaki, Miyazaki, Japan. ✉email: akira_sawaguchi@med.miyazaki-u.ac.jp

One of the major goals of morphological biology is to identify correlations between cell/tissue structures and functions. Visualization of the exact location of targeting molecules presents unique spatial information that no other biomedical analysis can provide. This process is called immunocytochemistry, a combination of immunochemistry and cell morphology, which uses specific antibodies against the target molecules of interest. In the last decade, many cell biologists have used immunocytochemistry, which has the benefits of improved fluorescence light microscopy, such as two-photon excitation microscopy; however, owing to the limited spatial resolution of light microscopy, electron microscopy is often needed to address important questions[1]. To close the gap between light and electron microscopy, researchers have recently used correlative light and electron microscopy (CLEM) to determine the orientation of complex cell/tissue architectures by light microscopy and identify ultrastructural correlations through electron microscopy[2].

Nanogold particles with diameters of 1–100 nm are also known as colloidal gold when dispersed in water. In 1857, Faraday first provided a scientific description of the optical properties of nanometer-scale gold synthesized by reducing chloroauric acid solution[3]. Since then, various experimental methods have been developed for the synthesis of nanogold particles based on Turkevich's pioneering study[4] in 1951, which identified the nucleation and growth process in the synthesis of colloidal gold. Recently, nanogold particles have attracted increasing interest because of their potential in nanotechnology-based on their unique size and three-dimensional structure[5,6].

In 1959, Singer[7] first reported an antibody tagged with the electron-dense protein ferritin for immunocytochemistry because the electron-scattering ability of fluorescent probes is insufficient for electron microscopy. Since the report by Faulk and Taylor in 1971[8], the general strategy for electron microscopic localization is the conjugation of nanogold "probes" to antibodies[9,10]. The nanogold probes are easily distinguished on the labeled cell/tissue structures and can be counted for quantitative analysis of labeling intensity. However, conventional methods require the time-consuming preparation of nanogold probes[11,12] and their conjugation to antibodies[9,10]. A variety of nanogold-conjugated antibodies are commercially available to decrease preparation, but ultrathin sectioning of the cell/tissue samples still requires time and skill for post-embedding nanogold probe labeling.

In a previous study[13], we introduced an informative three-dimensional survey of complex cell/tissue architectures using low-vacuum scanning electron microscopy (SEM) accompanied by CLEM imaging. This study aimed to develop its application in immunocytochemical localization, which remains challenging owing to difficulties in achieving labeling intensity with large particles. There is a trade-off relation between large particles and labeling intensity in post-embedding nanogold probe labeling[14], but large particles (>30 nm) are indispensable for visualization under low-vacuum SEM. In this context, we introduce a new strategy involving in situ nanogold nucleation in immunoenzymatic 3,3'-diaminobenzidine tetrahydrochloride (DAB) products on universal paraffin/cryostat sections and provide novel insight into nanogold development in hot-humid air conditions for ultrastructural localization.

## Results and discussion
### Practical in situ nanogold labeling
The proposed procedure employs the catalytic properties of an enzyme (horseradish peroxidase; HRP) that yields a red/brown-colored reaction product from DAB[15]. Compared with immunofluorescence, the enzyme system[16] is advantageous in that the resultant precipitates remain permanent and visible under a standard bright-field light microscope. Fig 1 illustrates the flow diagram from the light microscopic survey to the correlative ultrastructural localization of the target molecules (in this experiment, synaptopodin[17], expressed in the process of podocytes in the kidney glomerulus). After the light microscopic survey and preselection, the sections were treated with 0.01% tetrachloroauric acid (HAuCl$_4$) for 10 min, leading to nanogold particle "nucleation" (Fig. 1a). Then, the sections were exposed to hot-humid air in a humid chamber at 37 °C for 9–15 h to accomplish the "secondary growth" of the nucleated nanogold particles around the target molecules.

For electron microscopic observation, one section was placed on the wide specimen stage of the low-vacuum SEM (Fig. 1b), which was suitable for the whole-section survey at the centimeter scale (larger than the limited millimeter scale for conventional electron microscopy). Low-vacuum SEM allows backscattered electron imaging of non-conductive biological samples because the negative charge accumulations on the non-conductive materials can be neutralized by the positive ions in residual gas molecules[18]. As shown in the representative electron micrographs, nanogold particles were specifically localized on the process of podocytes in the kidney glomerulus, consistent with the preselected light microscopic findings. The underlying fine structures of the podocytes and their processes, which were not observable with conventional light microscopy, were clearly observed at higher magnification.

### Fortuitous nanogold development in a summer
The present in situ nanogold labeling was fortuitously developed on a laboratory bench, independent of previous studies. At the beginning of this study, we initially attempted to "enhance" the contrast of DAB deposition sites with 0.1% HAuCl$_4$ solution to convert the color signal into an electron-dense compound for electron microscopy[19,20]. It has been reported that exposure to gold chloride intensifies the electron density of DAB deposition sites[19]. In our early trials, however, the enhanced signals were indistinguishable from the enhanced electron emission, known as the edge effect, generated along the sectioned cell/tissue edges (Fig. 2a). The disappointing section was left on our laboratory bench over a hot-humid summer weekend. As a result, the section color unexpectedly changed from yellow to purple, indicating the development of nanogold particles[3]. Further experiments demonstrated a constant color change (Fig. 2b) and indicated nanogold labeling (Fig. 2c) accompanied by undesirable non-specific particles owing to an unrefined preliminary protocol.

Historically, the protocol for newly synthesized nanogold particles has been developed by extensive trial-and-error strategies rather than directed design[4–6]. In this study, the nanogold labeling efficiency and specificity were optimized by coordinating the HAuCl$_4$ concentration, treatment time, and subsequent hot-humid incubation. First, the optimal combination was determined in the range of 0.02–0.01% HAuCl$_4$ and 3–20 min by first performing hot-humid development at 37 °C for 12 h (Fig. 2d–g). Next, the optimal hot-humid development time was identified in the range of 9–15 h for standardized treatment with 0.01% HAuCl$_4$ for 10 min (Fig. 3). Finally, the optimal temperature was determined to be 37 °C for hot-humid development by the elimination of unsatisfactory development at 18 °C and morphological destruction at 60 °C (Fig. 3). Indispensably, the cell/tissue structures must be visible under ideal nanogold labeling to enable exact localization.

### In situ nanogold nucleation on the target site
Transmission electron microscopy (TEM) was applied to precisely localize the nanogold particles and define their morphological features at

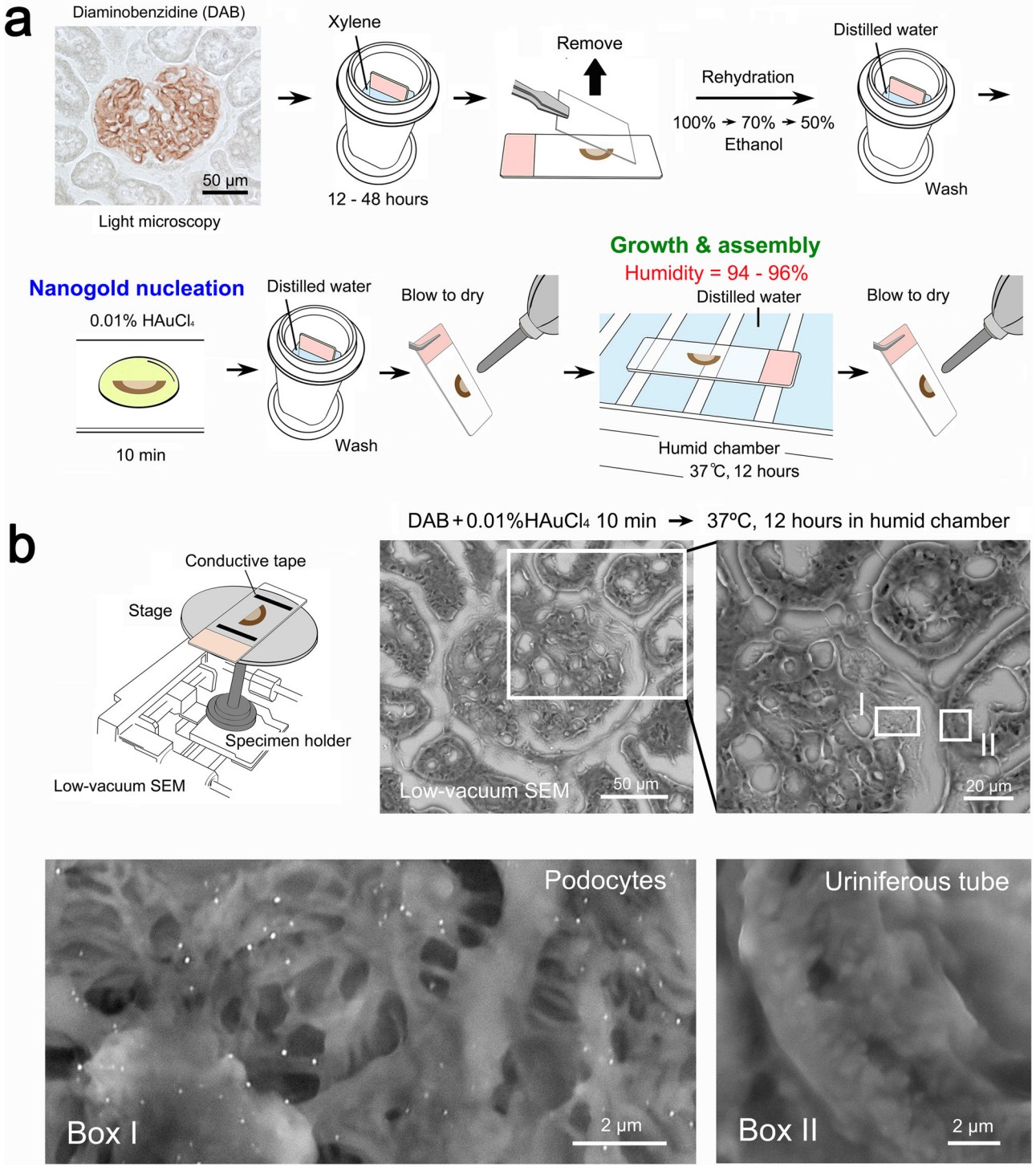

**Fig. 1 Practical flow diagram of in situ nanogold labeling and representative electron micrographs. a** First, the DAB deposition sites for the target molecules were surveyed under a light microscope. Then, the preselected microscope slides were incubated in xylene for 12–48 h to remove the coverslips and then rehydrated through an ethanol series. After the sections were washed in distilled water, they were treated with 0.01% HAuCl$_4$ for 10 min and then incubated in a humid chamber (with distilled water [DW] on the floor) at 37 °C for 12 h. **b** One microscope slide was placed on the wide stage of the specimen holder for low-vacuum SEM. The electron micrographs showed specific labeling on the processes of podocytes in the kidney glomerulus (in Box I), demonstrated by correlative light and electron microscopy (compare to the light micrograph in **a**). The cuboidal epithelium of the uriniferous tube, as a negative control, shows no particles (in Box II).

high resolution. The nanogold particles developed in situ were specifically localized in the processes of podocytes (Fig. 4a), consistent with the original report by conventional immunogold labeling[17]. The underlying processes of nanogold particle formation could be explained by LaMer's classic theory[21] and its

modifications, which have described the concept of burst nucleation as the first step in a phase transition. Consequently, the nucleated particles act as "seeds" for the secondary growth induced by a combination of monomer addition, aggregation, and coalescence.

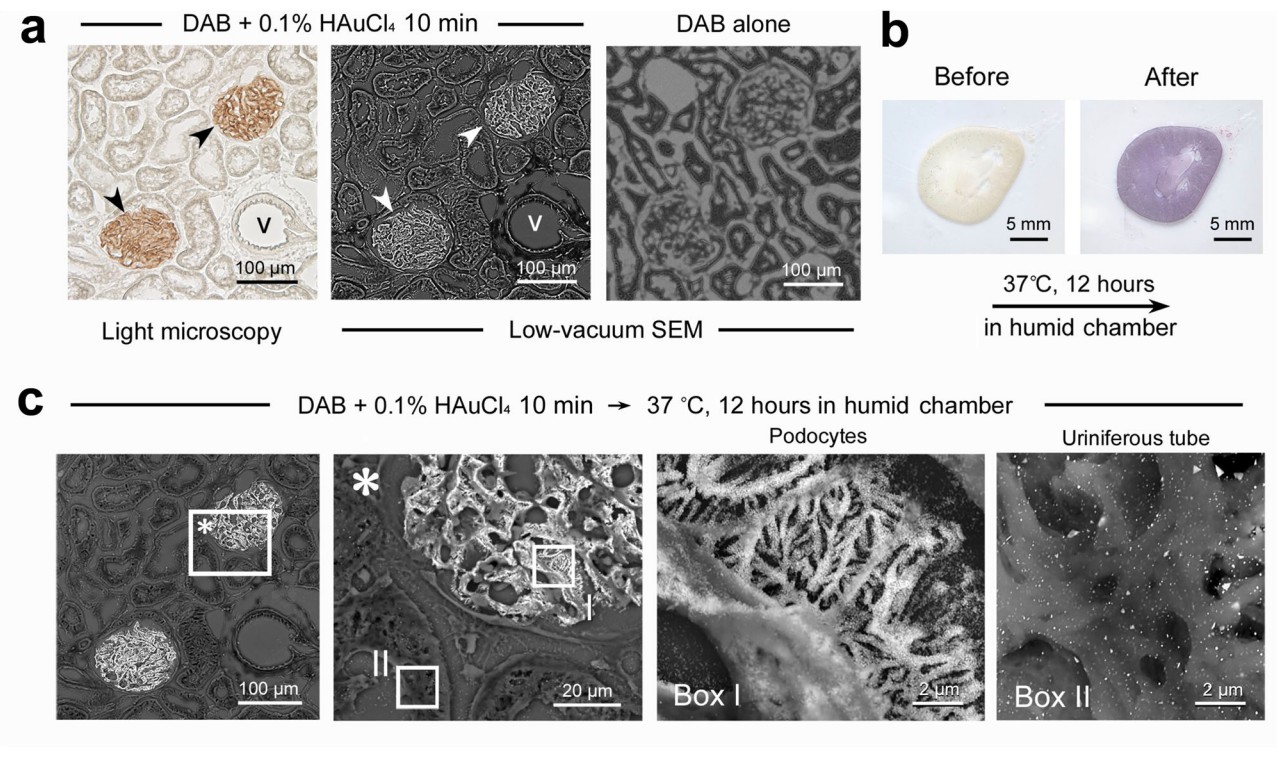

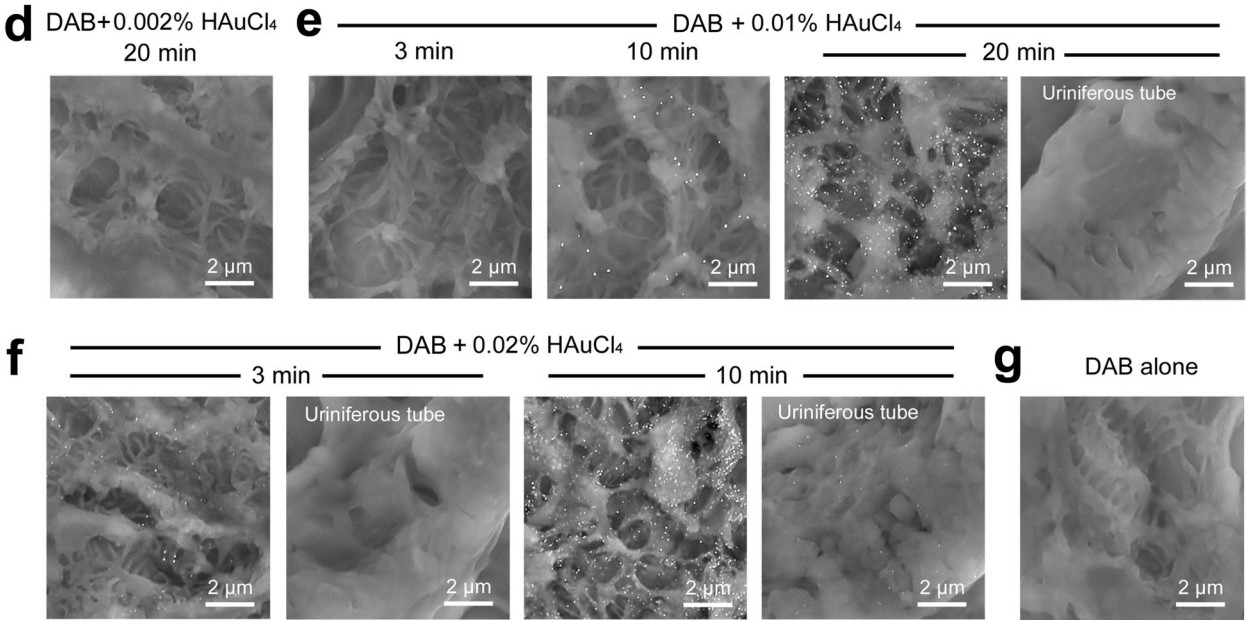

**Fig. 2 Prototype of the in situ nanogold labeling and coordination of HAuCl₄ concentration and treatment time. a** Contrast enhancement with 0.1% HAuCl$_4$ for the DAB deposition sites (arrowheads). Note the undifferentiated enhancement at the contours of the uriniferous tube and the blood vessel (v) caused by the edge effect under low-vacuum SEM. **b** Optical color change in the cell/tissue section after hot-humid incubation at 37 °C for 12 h. **c** Low-vacuum SEM images of the color-changed section. Note the intense nanogold labeling on the DAB deposition sites (in Box I) and nonspecific particles over the uriniferous tube (in Box II). **d–g** All sections were preset to be incubated in hot-humid conditions at 37 °C for 12 h. **d, e** Nanogold labeling was insufficient after treatment with 0.002% HAuCl$_4$ for 20 min (**d**) and 0.01% HAuCl$_4$ for 3 min compared with the standard treatment with 0.01% HAuCl$_4$ for 10 min (**e**). Intense labeling was obtained with 0.01% HAuCl$_4$ for 20 min, but it obscured the underlying fine structure of the podocyte process. **f** Appropriate labeling was obtained with 0.02% HAuCl$_4$ for 3 min, but the short elongation of 10 min caused nonspecific particles on the uriniferous tube. **g** DAB alone as a negative control.

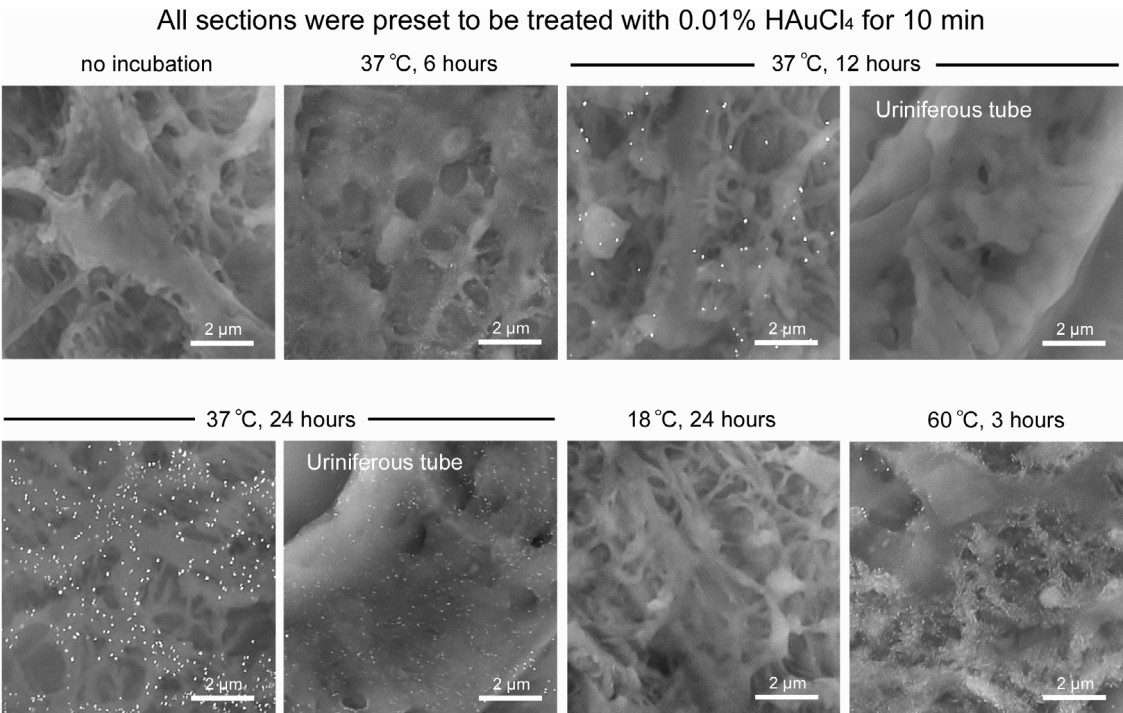

**Fig. 3 Coordination of the duration and temperature of hot-humid incubation.** All sections were preset to be treated with 0.01% HAuCl₄ for 10 min. Note the immature growth of nanogold particles after hot-humid incubation at 37 °C for 6 h. Satisfactory labeling was observed after 12 h, but elongation to 24 h caused nonspecific particles on the uriniferous tube. No labeling was developed in the humid chamber at 18 °C for 24 h. Incubation at 60 °C accelerated nanogold development, but the underlying structure of the podocyte process was significantly damaged.

In this study, the primary step of nanogold particle "nucleation" was confirmed among the immunoenzymatic DAB products for the target molecules after treatment with 0.01% HAuCl₄ alone for 10 min (Fig. 4b). The speculated reactivity between HAuCl₄ and DAB was verified by dot blotting on filter paper[20] (Fig. 4c) and by dripping 0.05% HAuCl₄ into 0.02% DAB aqueous solution, which produced brown grains in a microtube. In immunohistochemical staining, DAB is known to be oxidized by hydrogen peroxide (H₂O₂) in the presence of HRP that forms a brown deposition, representing the location of the HRP for light microscopy. Intensifications of DAB deposition sites have been reported with various heavy metallic ions as well as gold chloride[19,20,22–25]. The crucial binding ability between gold chloride and the immunochemical reaction product of DAB has been indicated by energy dispersive X (EDX)-ray analysis[25]. Interestingly, oxidative polymerization of DAB on gold electrode has been reported in an electrochemical study for the preparation of polymeric film coated electrode[26]. However, further analysis is needed due to the lack of detailed knowledge concerning DAB polymerization and the chemical characteristics of the resultant deposition in immunohistochemistry.

**Secondary growth in hot-humid air conditions.** An advanced series of TEM images demonstrated the secondary growth of nanogold particles via seed-mediated "monomer addition", "aggregation", and "coalescence" in accordance with LaMer's theory[4,5,21,27] (Fig. 4d). The secondary growth resulted in a significant increase in the average diameter (Fig. 4e) accompanied by a broadening of the size distribution (Fig. 4f). Concerning the gold source for monomer addition, elemental analysis indicated that the sections were equally stained with HAuCl₄ regardless of the DAB deposition sites (Fig. 5a). It could be assumed that gold monomers were supplied from the overall staining to the nanogold particles by liquid diffusion because the microscope slides

were moistened after the hot-humid incubation. In fact, the secondary growth failed in incubation in dry air, which might lack the dissolved gold monomers, and in waterdrop incubation, which might dilute the dissolved gold monomers (Fig. 5b).

Further TEM revealed the changes in nanogold configurations from spherical to multibranched polynuclear assemblies (alternatively referred to as gold nanostars[28] and nanoflowers[29]) and the associated merged lines[30] (Fig. 6a). High-resolution scanning transmission electron microscopy (HR-STEM) demonstrated representative bright-field and annular dark-field images of intense nanogold labeling on the processes of podocytes (Fig. 6b). Higher magnification clearly shows the details of multiple crystalline structures in the attachment region (Fig. 6c) and lattice fringes with interplanar spacing of 0.24 nm corresponding to Au (111) planes accompanied by fast Fourier transform (FFT) patterns (Fig. 6d). Recently, studies of aggregating motion have been advanced by in situ liquid-cell TEM nanotechnologies that enable the "real-time" observation of aggregating nanogold particle motion in a cluster[31–34] and manipulation of individual particles by electron beam[35]. It will be of great interest to apply such forefront technologies to elucidate the precise mechanism of secondary growth in hot-humid air conditions, reaching large-sized intense labeling in a short time.

**Influence of pH on nanogold particle size.** Nanogold particles are unstable and tend to aggregate at short interparticle distances, which has attracted increased attention in the field of colloid science. Notably, secondary growth was reproduced on a dot blotting spot by hot-humid incubation at 37 °C for 12 h (Fig. 6e), providing a new filter paper assay for this study. In this protocol, the cell/tissue sections were treated with H₂O₂ as usual in enzyme-based immunocytochemistry. It could be speculated that residual H₂O₂ readily reduced HAuCl₄ and accelerated secondary growth[36–39]. However, this speculation was contradicted by the

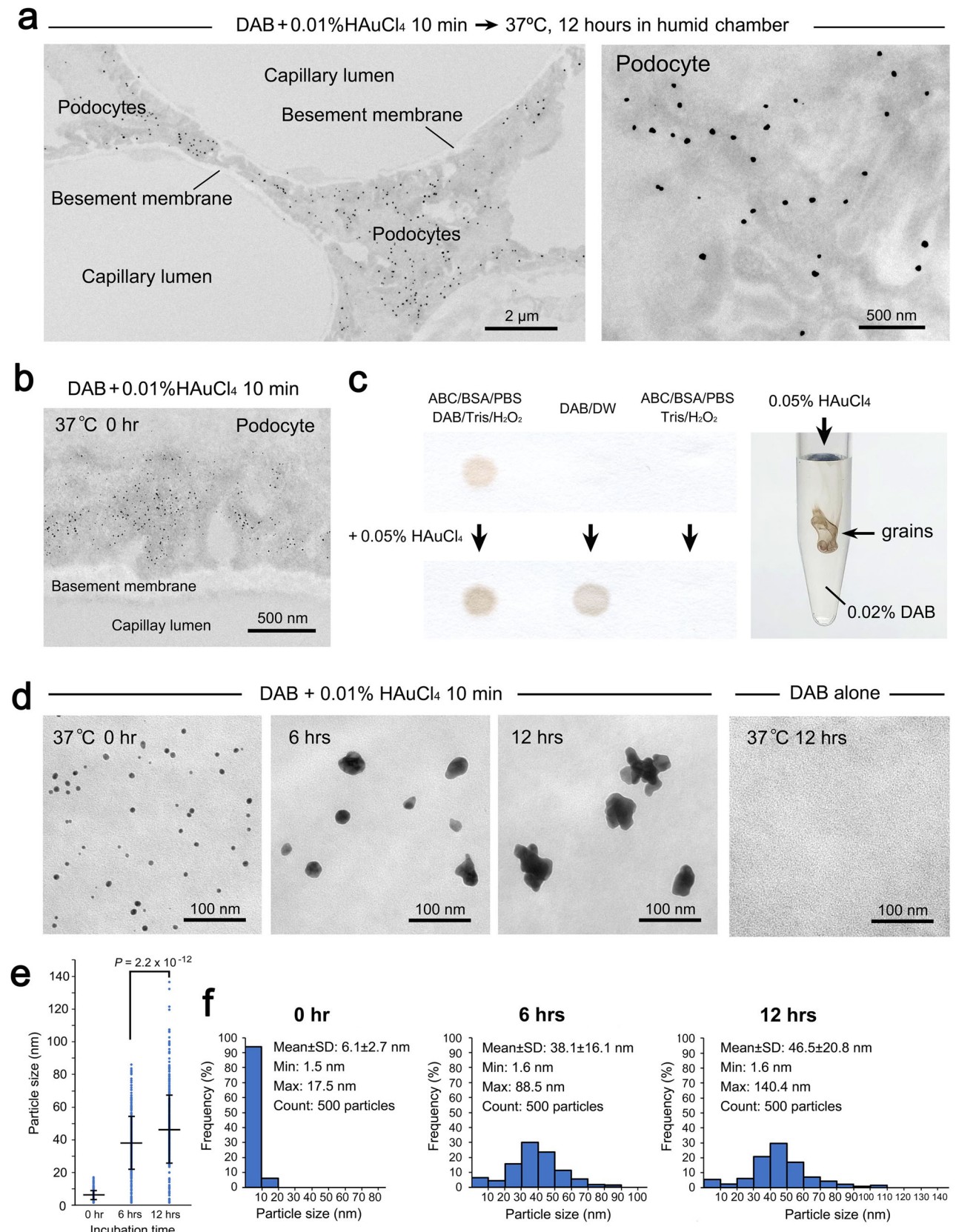

nucleation and secondary growth observed in $H_2O_2$-free blotting spots.

It is well known that the chemical properties of $HAuCl_4$ solution depend on its pH[26,39–41]. The primary screening by filter paper model assay showed a decline in the purplish tone in proportion to increasing pH value (Fig. 7a). Correspondingly, the low-vacuum SEM (Fig. 7b) demonstrated the reduction in labeling intensity on the sections. Further TEM observations (Fig. 7c) and quantitative analysis (Fig. 7d, e) clarified the reduction in secondary growth, which could be explained by the

**Fig. 4 TEM analyses of nanogold nucleation and secondary growth. a** Ultrathin TEM images of the in situ nanogold labeling of synaptopodin treated with 0.01% HAuCl$_4$ for 10 min followed by hot-humid incubation at 37 °C for 12 h. Note the specific labeling on the processes of podocytes. **b** Nucleated nanogold particles after treatment with 0.01% HAuCl$_4$ for 10 min. **c** Visualization of the chemical reaction between HAuCl$_4$ and DAB on dot blots and in vitro. *ABC* avidin-biotin complex, *BSA* bovine serum albumin, *PBS* phosphate-buffered saline, *DAB* diaminobenzidine, *DW* distilled water. **d** Time-lapse growth and changes from spherical to multibranched shapes. **e, f** Quantitative comparison of the average diameter (**e**) and the size distribution (**f**) of the nanogold particles. *n* = 500 particles/group. All data are expressed as the standard deviation (SD) ± mean. Statistical significance was assessed by Student's *t* test.

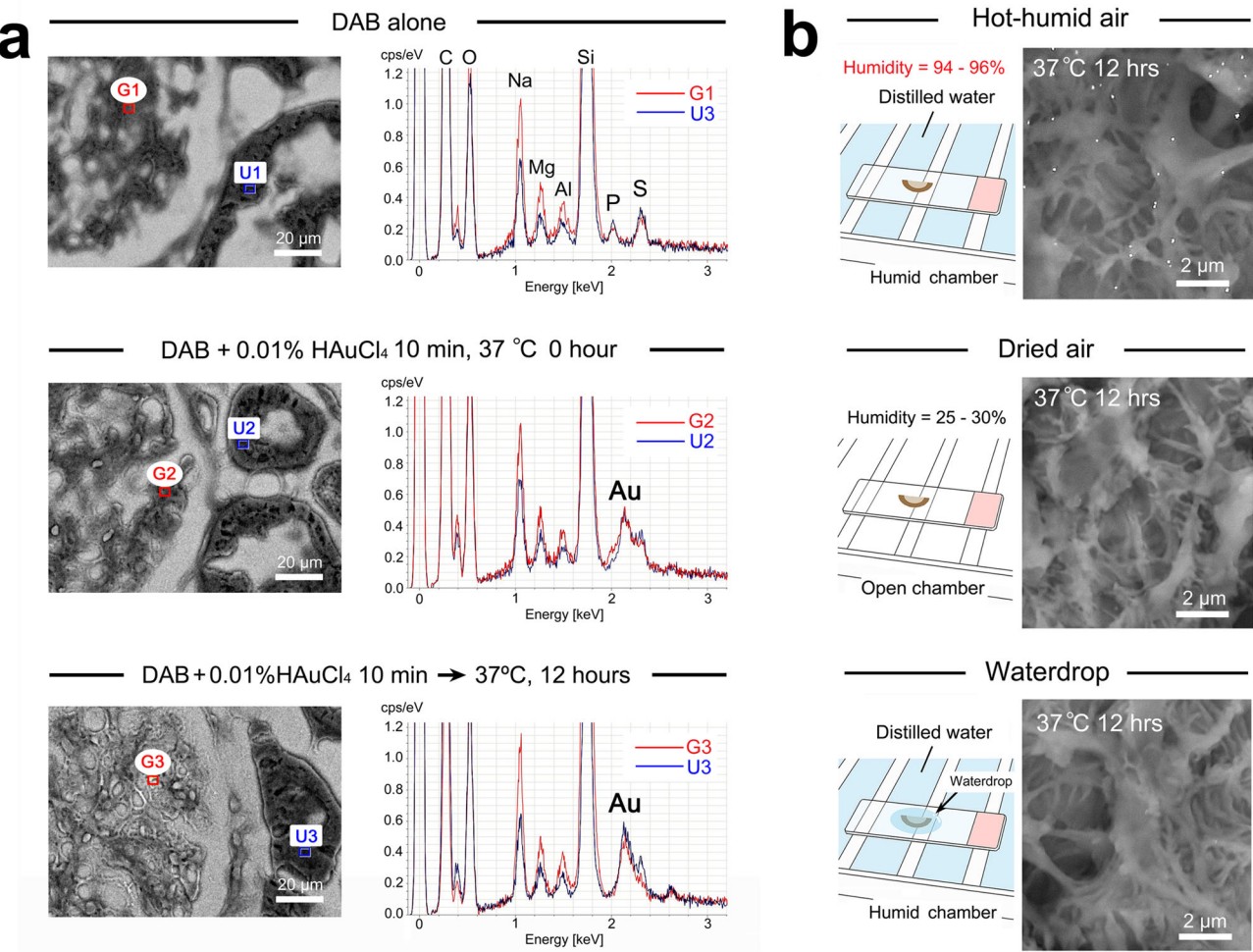

**Fig. 5 Elemental analysis of Au distribution in the HAuCl$_4$-treated sections and conditioning for secondary growth. a** A clear Au peak emerged after treatment with 0.01% HAuCl$_4$ for 10 min compared with negative control of DAB alone. The Au peak levels were irrelevant to DAB deposition and equally observed in the glomerulus (G2) and uriniferous tube (U2). Note the unchanged Au peak pattern at G3 and U3 after hot-humid incubation at 37 °C for 12 h. G: glomerulus. *U* uriniferous tube. **b** Conditioning for secondary growth. Distinct nanogold particles were developed in high-humidity air but not dried air or waterdrops and were incubated at 37 °C for 12 h.

stabilization of nanogold particles at higher pH, consistent with previous reports[40–42].

**Experimental control of aggregation.** Nanogold particles can be stabilized when macromolecules are adsorbed on the surface, creating a mechanical barrier against aggregation. Practically, bovine serum albumin (BSA) is added to solutions of antibody-conjugated nanogold particles as a "capping" agent in conventional methods[10,12]. A filter paper assay showed a decline in the purplish tone after pretreatment with 2% BSA for 10 min prior to hot-humid incubation, indicating suppression of the secondary growth (Fig. 8a). Correspondingly, stabilization was proven on cell/tissue sections (Fig. 8b, c, e, f). HR-STEM revealed multiple

crystalline structures within the spherical particles (Fig. 8c), implicating polynuclear coalescence under suppression.

Successful stabilization has increased interest in destabilization by electrolytes that induce aggregation by reducing electrostatic repulsion[43]. Indeed, pretreatment with phosphate-buffered saline (PBS) accelerated the growth and aggregation of nanogold particles, leading to massive polynuclear configurations (Fig. 8d, e, g). This series of experiments and findings will play an important role in verifying theoretical models for the process of nanogold particle development.

**Potential in applied immunocytochemistry.** Fig 9a shows a diagram of the nanogold particle labeling originating from the in situ nucleation of immunoenzymatic DAB products, followed

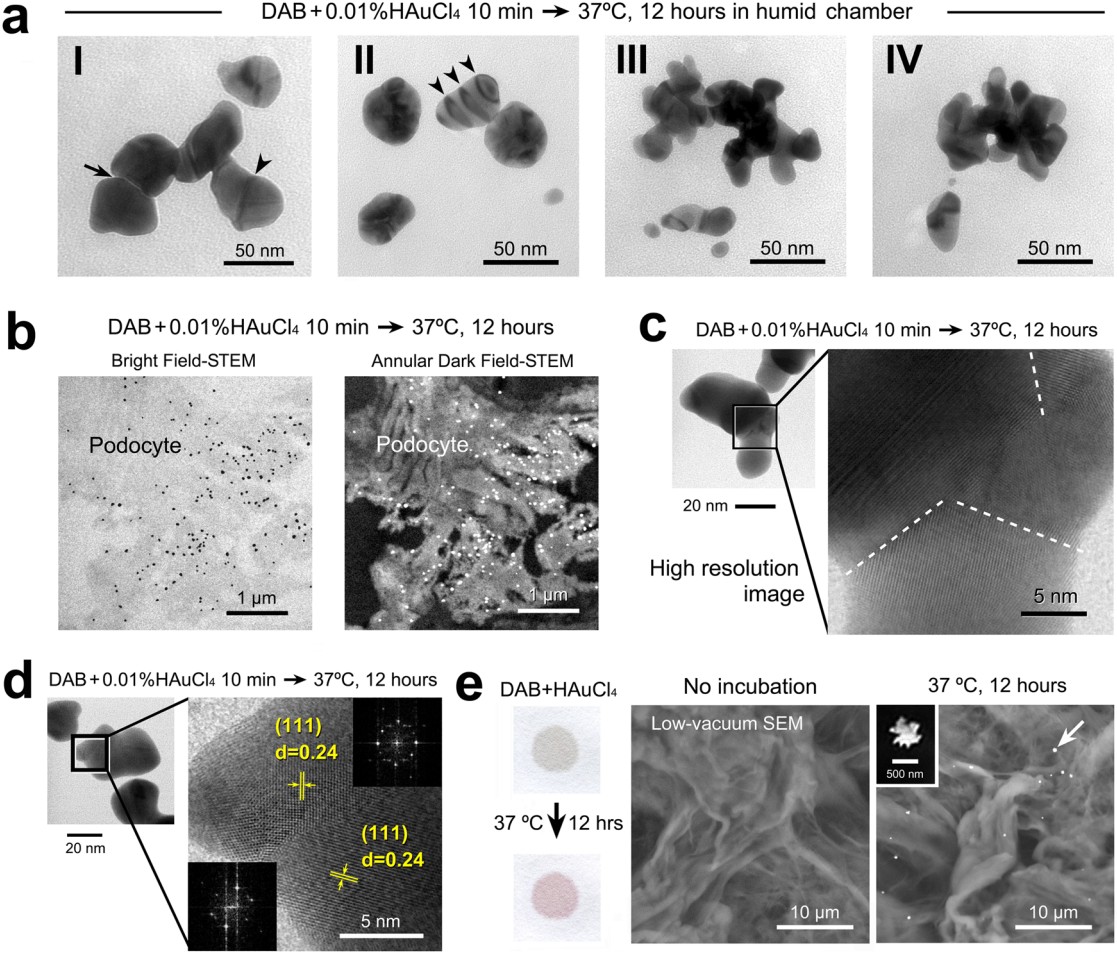

**Fig. 6 The secondary growth and aggregation of nanogold particles. a** Characteristic nanogold assemblies after hot-humid incubation at 37 °C for 12 h. **a–I** A representative branched feature of the nanogold particles. Note the attached portion (arrow) and merged line (arrowhead) within the assembly. **a–II** Heterogenous light and shade patterns are shown in each nanogold particle. Arrowheads indicate the stripe pattern at regular intervals. **a–III** Multibranched assembly. **a–IV** A multibranched assembly surrounding a hole. **b–d** HR-STEM images. Representative bright-field and annular dark-field images of intense nanogold labeling on the processes of podocytes (**b**). Higher magnification clearly shows the details of multiple crystalline structures in the attachment region (**c**) and lattice fringes (**d**) with interplanar spacing and FFT patterns. **e** A dot blotting spot exhibited a color change similar to that of cell/tissue sections after hot-humid air incubation at 37 °C for 12 h. The low-vacuum SEM micrographs show secondary growth on the filter paper with a representative "star-like" nanogold particle (inset).

by secondary growth and aggregation under hot-humid air conditions. This in situ nanogold labeling employed the catalytic properties of an enzyme that yields semipermanent precipitates on the target molecules. By taking advantage of this process, we achieved retrospective nanogold labeling of gastric H⁺/K⁺-ATPase on vintage DAB deposits after a long lapse of 15 years[44] (i.e., 15-year-old-DAB deposits, Fig. 9b; cf. Fig 3C in ref. [44]). The paraffin and cryostat blocks of cell/tissue samples are semipermanent and show potential in retrospective investigations by the reappraisal of archived samples. The pathological application showed intense labeling of platelet glycoprotein IIb/IIIa preserved in an archived specimen of rabbit arterial thrombosis[45] (Fig. 9c). Moreover, this method can be combined with in situ hybridization, which enables the ultrastructural localization of a defined sequence of DNA or RNA on cell/tissue sections[46] (Fig. 10).

Recent advances in biomedical research, such as the production of regenerative organs from induced pluripotent stem cells and the morphological changes induced by CRISPR/Cas9-mediated genome editing, require the ultrastructural localization of a particular molecule to correlate cell/tissue structure and function. The present in situ nanogold labeling is user-friendly with basic paraffin/cryostat sections and highly anticipated to create a new approach in applied biomedical immunocytochemistry, bridging the gap between light and electron microscopy.

## Methods

**Preparation of rat kidney cryostat sections.** Male Wistar rats (Kyudo, Kumamoto, Japan) at 10 weeks of age were deeply anesthetized and then perfused with 4% paraformaldehyde in 0.1 M phosphate buffer (pH 7.4) from the left ventricle of the heart. The kidney was excised and further fixed by immersion in the above fixative for 2 h at room temperature (RT). After the organs were washed in running tap water for 2 h, they were protected in a graded series of 5%, 10 and 20% sucrose in PBS for 2 h and in a mixture of 1:1 (v/v) 20% sucrose:OCT compound (Sakura Finetek, Tokyo, Japan) at 4 °C overnight. The samples were then embedded in OCT compound and cut into sections (10 μm in thickness) with a cryostat (CM3050 S, Leica Microsystems, Wetzlar, Germany).

**Immunocytochemistry of synaptopodin.** The cryostat sections of rat kidney were rinsed in PBS and then boiled in 10 mM citrate buffer, pH 6.0, for 10 min to retrieve the antigenic sites. After cooling to RT, the sections were incubated in methanol containing 0.3% H₂O₂ for 30 min to block endogenous peroxidase activity. After several rinses in PBS, the sections were incubated in 5% normal horse serum (NHS)/1% BSA in PBS for 10 min to block nonspecific binding and then incubated with a mouse monoclonal antibody against synaptopodin (clone G1D4; Progen Biotechnik, Heidelberg, Germany; diluted 1:50 with 5% NHS/1% BSA in PBS) at 4 °C overnight. As controls, the procedure was performed without

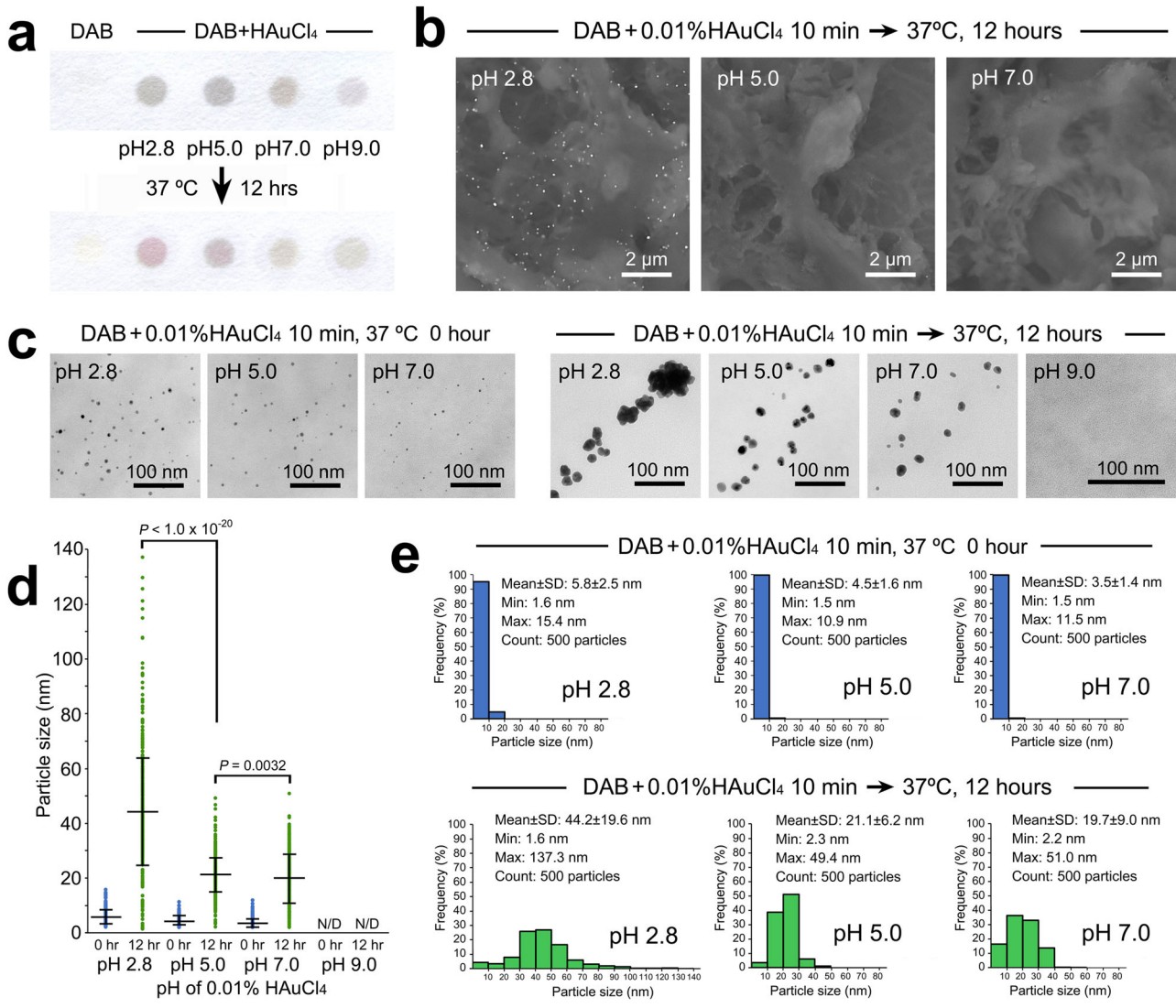

**Fig. 7 Influence of pH on nanogold size. a** Primary screening by filter paper model assay. Note the decline in the purplish tone in proportion to the increasing pH value of HAuCl₄ solutions. **b** Low-vacuum SEM. Note the satisfactory labeling at pH 2.8 and undistinguishable intensities at pH 5.0 and pH 7.0 after hot-humid incubation at 37 °C for 12 h. **c** Ultrathin TEM images of the nanogold particles before and after hot-humid incubation at 37 °C for 12 h. **d**, **e** Quantitative comparison of the average diameter (**d**) and size distribution (**e**) of the nanogold particles. $n = 500$ particles/group. All data are expressed as the standard deviation (SD) ± mean. Statistical significance was assessed by Student's $t$ test.

primary antibodies. After the sections were washed with PBS, they were incubated with biotinylated horse anti-mouse IgG (Vector Laboratories, Burlingame, CA, USA; diluted 1:200 with 1% BSA in PBS) at RT for 40 min, followed by washing with PBS. Then, sections were incubated in a freshly prepared solution of avidin-biotinylated HRP complex (ABC) kit (Vector Laboratories; diluted 1:50 with 1% BSA in PBS) for 30 min. After PBS washes, the peroxidase reaction was developed by incubating in 0.05% DAB (Dojindo Laboratories, Kumamoto, Japan) in 0.05 M Tris buffer, pH 7.6, containing 0.001% H₂O₂ for 8 min.

**Light microscopic survey**. For the light microscopic survey of the DAB deposition sites, the sections were rinsed in a graded series of 80%, 90%, and 100% ethanol for dehydration. After clearance in xylene, the sections were mounted within Malinol (Muto Pure Chemicals, Tokyo, Japan) covered with a NEO Micro cover glass (size 24 × 50 mm, thickness no. 1 = 0.13–0.17 mm: Matsunami Glass, Osaka, Japan). The DAB deposition sites were surveyed under a light microscope (BX51, Olympus, Tokyo, Japan) equipped with a digital camera (DP73, Olympus).

**Nanogold particle development**. Next, for light microscopy, the microscope slides were incubated in xylene for 18–48 h to remove the coverslips by dissolving the fixed mounting medium. Then, the sections were rehydrated with a series of 100%, 90%, and 70% ethanol (5 min each). After several rinses in distilled water (DW; at least three times, 5 min each), the sections were treated with a 0.01% aqueous

solution of hydrogen tetrachloroaurate (III) tetrahydrate (HAuCl₄·4H₂O, Nakarai Tesque, Kyoto, Japan) for 10 min at RT. After being rinsed several times in DW and dried, the sections were incubated in a humid chamber (equipped with DW on the floor, adequate humidity = 94–96%) at 37 °C for 12 h and then dried with a blower. As controls, other sections were incubated in dried air (humidity = 25–30%) or waterdrops.

The above optimal protocol was determined by the coordination of HAuCl₄ concentration (0.001, 0.002, 0.005, 0.01, 0.02, or 0.1%), HAuCl₄ treatment time (3, 5, 10, or 20 min), subsequent incubation time in a humid chamber (0, 3, 6, 9, 12, 15, 18, or 24 h) and temperature (18, 37, or 60 °C). The influence of the pH value on nanogold particle development was examined by changing the non-adjusted initial pH value of 0.01% HAuCl₄/DW (pH 2.8) to pH 5.0, pH 7.0, and pH 9.0 adjusted with 0.2 M K₂CO₃. The physicochemical properties of the nanogold particles were examined by treatment with 2% BSA/DW for 10 min or PBS for 10 min, followed by several rinses with DW prior to incubation in a humid chamber.

**Low-vacuum SEM**. The microscope slides were placed on the wide stage of the specimen holder using adhesive conductive tape and then placed in a low-vacuum SEM (TM4000Plus or TM3030Plus, Hitachi High-Tech, Tokyo, Japan) operating at 15 kV. Elemental analysis was performed by using an EDX detector equipped for low-vacuum SEM.

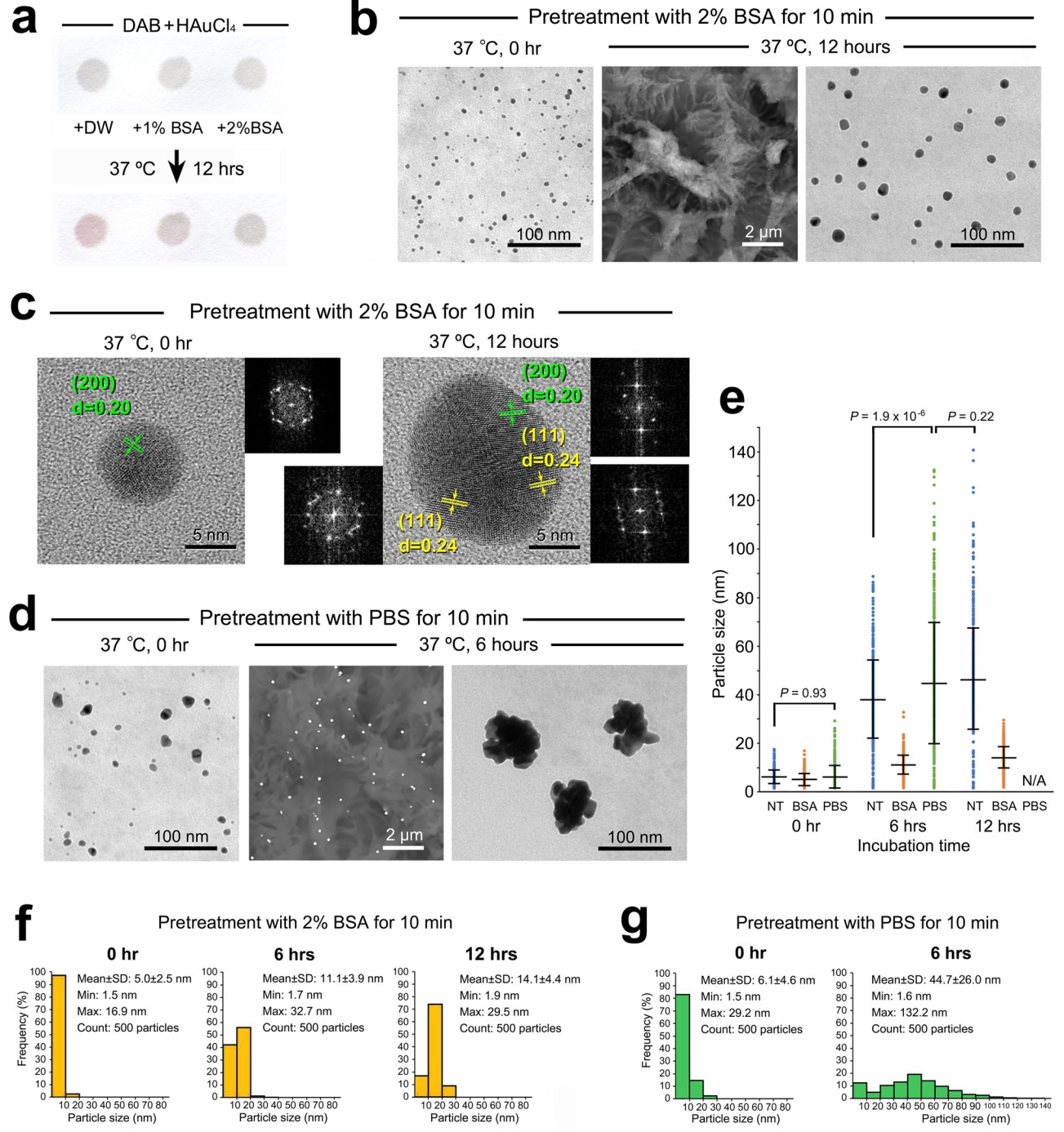

**Fig. 8 Inhibition and acceleration of secondary growth after pretreatment with BSA and PBS. a** Filter paper model assay for nanogold stabilization. Note the decline in the color change induced by pretreatment with 2% BSA for 10 min prior to hot-humid incubation. **b** Ultrathin TEM images of the nanogold particles after pretreatment with 2% BSA for 10 min prior to hot-humid incubation. Note that the secondary growth remained in a spherical form rather than a polynuclear configuration. **c** HR-STEM images of lattice fringes with interplanar spacing and FFT patterns. Note the multiple crystalline structures in the spherical particle after hot-humid incubation. **d** Pretreatment with PBS for 10 min accelerated growth and aggregation in polynuclear configurations. **e** Quantitative comparison of the average diameter. Because of the emergence of nonspecific particles, the average diameter was not applicable for 12 h of incubation after PBS pretreatment. *NT* not treated, *N/A* not applicable. **f** Size distribution of the nanogold particles after pretreatment with 2% BSA for 10 min prior to hot-humid incubation at 37 °C for 0, 6, and 12 h. **g** Size distribution after pretreatment with PBS for 10 min prior to hot-humid incubation at 37 °C for 0 and 6 h. Because of the emergence of nonspecific particles, the size distribution was not applicable for 12 h of incubation after PBS pretreatment. $n = 500$ particles/group. All data are expressed as the standard deviation (SD) ± mean. Statistical significance was assessed by Student's *t* test.

**Characterization of the developed nanogold particles**. The nanogold particles were morphologically characterized by using a transmission electron microscope. For the characterization of nanogold particles, the sections were dehydrated in a graded series of 80%, 90%, and 100% ethanol and embedded into epoxy resin on microscope slides by using a capsule-supporting ring[47]. Subsequently, the polymerized resin block was removed from the microscope slide by rapid cooling in liquid nitrogen vapor. Ultrathin sections (60–80 nm in thickness) were cut and observed without heavy metal staining by using TEM (HT7700, Hitachi-High Tech, Tokyo, Japan) operating at 80 kV or HR-STEM (HD-2700A, Hitachi-High Tech, Tokyo, Japan) operating at 200 kV.

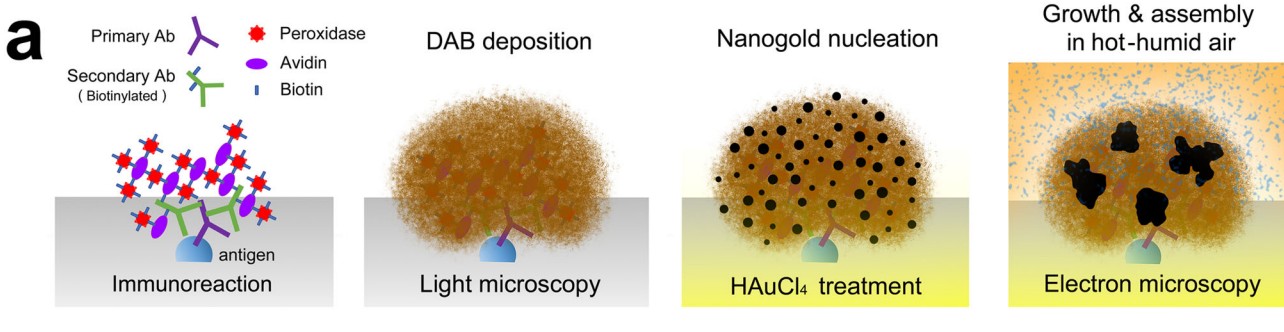

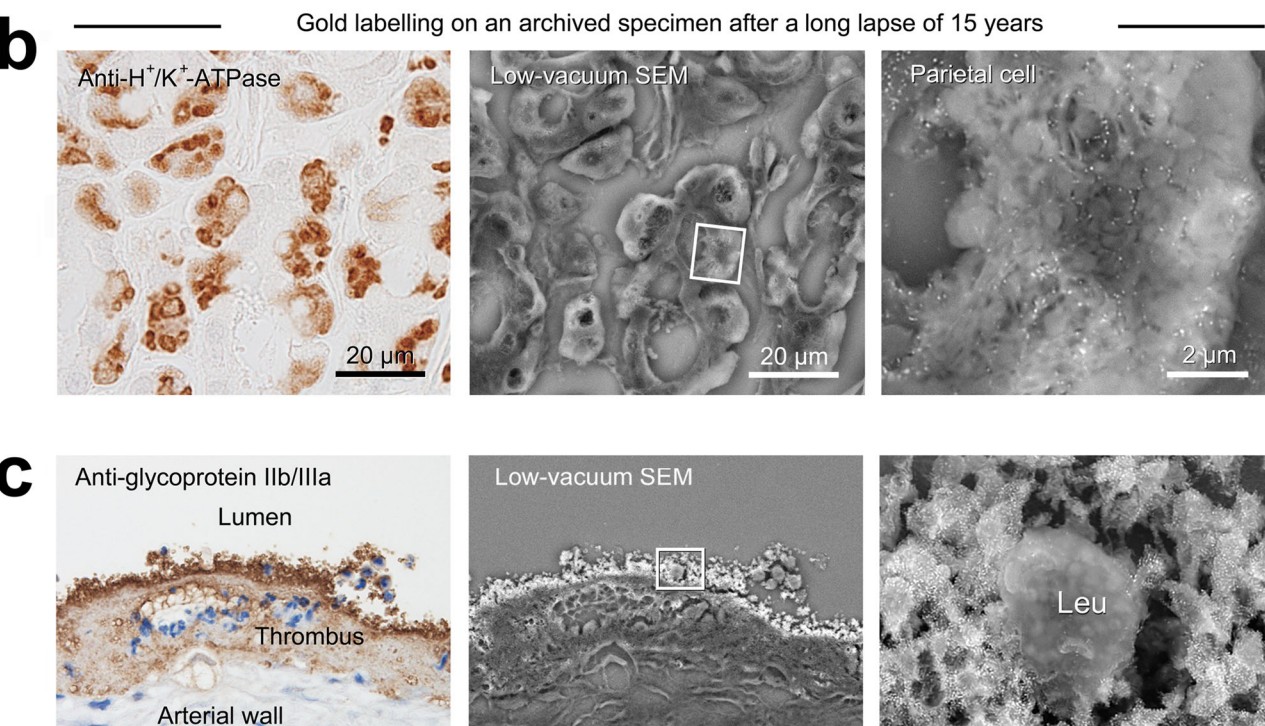

**Fig. 9 Illustration of in situ nanogold labeling and biomedical applications. a** Proposed illustration of the in situ nanogold labeling. Nanogold particles nucleated from HAuCl₄ in situ among the DAB deposition sites. The secondary growth and aggregation that occur in hot-humid conditions are sufficient for low-vacuum SEM observation. **b** Rat gastric parietal cell. Retrospective nanogold labeling of H⁺/K⁺-ATPase on a valuable archived specimen after a long lapse of 15 years (i.e., 15 years old-DAB depositions). **c** Pathological application to rabbit arterial thrombosis. Note the intense labeling of glycoprotein IIb/IIIa expressed on the platelets surrounding the leukocyte (Leu). The cell nuclei were stained with hematoxylin for light microscopy.

**Dot blot model analysis on filter paper**. The reactivity of HAuCl₄ to DAB was verified by dot blotting[19]. First, one dot (2 µl aliquot) of 0.05% DAB in DW (Fig. 4c, middle) and two dots of the ABC kit reagent diluted 1:50 with 1% BSA/PBS were blotted on filter paper (No. 131, Advantec, Tokyo, Japan) as controls. Next, the latter two controls of the ABC kit/1% BSA/PBS dots were blotted with 0.05% DAB/0.05 M Tris buffer, pH 7.6/0.001% H₂O₂ (Fig. 4c, left), or 0.05 M Tris buffer, pH 7.6/0.001% H₂O₂ (Fig. 4c, right). Subsequently, all three dots were blotted with 0.05% HAuCl₄/DW and air-dried.

The resultant brown dots of 0.05% DAB/DW and 0.05% HAuCl₄/DW were clipped out and mounted on microscope slides using adhesive conductive tape. The development of nanogold particles was examined in a humidity chamber at 37 °C for 12 h, as described above for the cell/tissue sections. The influence of pH on nanogold particle development was examined by adjusting the initial pH value of 0.01% HAuCl₄/DW (pH 2.8) to pH 5.0, pH 7.0, and pH 9.0 with 0.2 M K₂CO₃. The BSA stabilization test was performed with 1% or 2% BSA/DW by blotting over the brown dots of 0.05% DAB/DW and 0.05% HAuCl₄/DW prior to incubation in the humid chamber.

**Application to archived immunocytochemical specimens**. Archived paraffin sections of isolated gastric mucosa were employed from the immunocytochemical study by Sawaguchi et al.[44] (cf. Fig. 3C in ref. [44]). In brief, a piece of isolated gastric mucosa was cryofixed in a liquid isopentane/propane mixture cooled by liquid

nitrogen. Freeze substitution was carried out in 0.1% glutaraldehyde in acetone at −80 °C for 16 h, and the specimen was then gradually warmed to RT. After the specimens were washed with ethanol, they were embedded in paraffin. Paraffin sections (5 µm in thickness) were deparaffinized, rehydrated, and incubated in methanol containing 0.3% H₂O₂ for 30 min. After several rinses in PBS, the sections were incubated in 5% NHS/1% BSA in PBS for 10 min to block nonspecific binding and then incubated with a mouse monoclonal antibody, 2B6, against H⁺/K⁺-ATPase (MBL, Nagoya, Japan; 2 µg/ml diluted with 5% NHS/1% BSA in PBS) at 4 °C overnight. After PBS washes, the sections were incubated with biotinylated horse anti-mouse IgG at RT for 40 min, followed by washing with PBS. Then, the sections were incubated in an ABC kit for 30 min. The samples were washed again with PBS, and the peroxidase reaction was developed as described above.

In addition to light microscopy, after a long lapse of 15 years from the original preparation, the microscope slides were incubated in xylene for 48 h to remove the coverslips. Then, the sections were rehydrated with a series of 100%, 90%, and 70% ethanol. After several rinses in DW, nanogold particles were developed as described above.

**Application to pathological immunocytochemistry**. Paraffin sections of rabbit arterial thrombosis were obtained from a disturbed blood flow model[45]. In brief, the left femoral arteries of male Japanese white rabbits (Kyudo Corp, Kumamoto, Japan) weighing 2.5–3.0 kg were damaged by inserting a 2.5 (diameter) × 9 (length)

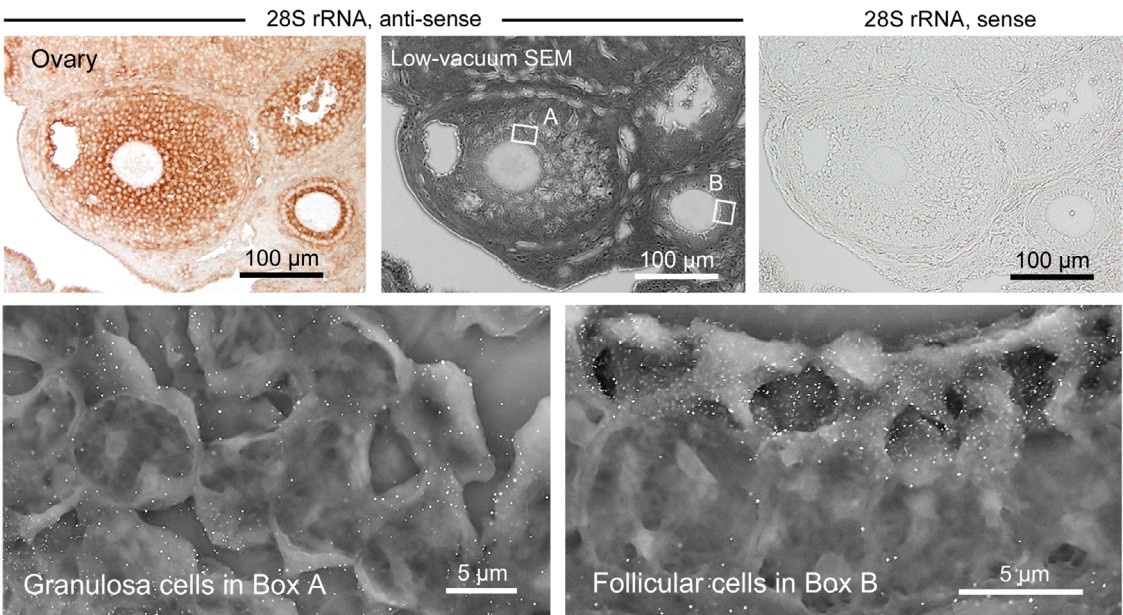

**Fig. 10 Application of CLEM to in situ hybridization for gene targeting.** CLEM was applied to in situ hybridization for gene targeting of 28 S rRNA on the mouse ovary. Note the intense nanogold labeling on the granulosa cells in Box A and the apical region of the follicular cells in Box B. The 28 S rRNA sense probe, as a negative control, showed no signal.

mm angioplasty balloon catheter (Boston Scientific, Galway, Ireland) into the femoral artery via the carotid artery. Three weeks later, the damaged femoral arteries were constricted to reduce the flow volume. Then, the rabbits were intravenously injected with heparin (500 U/kg) 15, 30, and 180 min thereafter, and then euthanized with an overdose of pentobarbital (60 mg/kg, intravenous) to evaluate thrombus formation. The animals were perfused with 4% paraformaldehyde, and the femoral artery was embedded in paraffin and sectioned (3 μm in thickness).

After deparaffinization, the sections were incubated in methanol containing 0.3% $H_2O_2$ for 30 min to block endogenous peroxidase activity. After several rinses in PBS, the sections were incubated in Protein Block (X0909; Agilent, Santa Clara, CA, USA) for 10 min to block nonspecific binding and then incubated with a sheep polyclonal antibody against the platelet glycoprotein IIb/IIIa (Affinity Biologicals, Inc., Hamilton, CA, USA; diluted 1:500 with Antibody Diluent purchased from Agilent) at 4 °C overnight. The procedure was performed without primary antibodies as a control. After the sections were washed with PBS, they were incubated with biotinylated donkey anti-sheep IgG (Jackson ImmunoReseach Laboratories, West Grove, PA, USA; diluted 1:1000 with Antibody Diluent) at RT for 30 min, followed by washing with PBS. Then, the sections were incubated in a ready-to-use solution of peroxidase-labeled streptavidin (Nichirei Biosciences, Tokyo, Japan) for 5 min. After the samples were washed with PBS, the peroxidase reaction was developed as described above. The cell nuclei were stained with Mayer's hematoxylin for 3 min and exposed to running tap water for at least 3 h to develop the color.

**Application to in situ hybridization**. Paraffin sections (5 μm in thickness) of the ovary from C57BL/6 J female mice, 8–12 weeks of age, were prepared by chemical fixation with 4% paraformaldehyde in PBS at RT for 24 h. After deparaffinization, the sections were treated with 0.2 N HCl and digested with proteinase K. After postfixation with 4% paraformaldehyde in PBS for 5 min, the sections were immersed in 2 mg/ml glycine in PBS for 30 min and kept in hybridization medium, 40% deionized formamide in 4× standard saline citrate (SSC) (1× SSC = 0.15 M sodium chloride and 0.015 M sodium citrate, pH 7.0), until hybridization. Hybridization was carried out at 37 °C overnight with digoxigenin-labeled oligo-DNAs for 28 S rRNA (2173–2206) dissolved in hybridization medium[46]. After repeated washings with 2× SSC, the sections were incubated with HRP-labeled sheep anti-digoxigenin antibody (Roche Diagnostics, Mannheim, Germany). After the samples were washed with PBS, the peroxidase reaction was developed as described above.

All animal procedures were carried out under protocols approved by the University of Miyazaki Animal Research Committee (#2005-009-1, #2016-513-3, #2016-509-5, #2018-509-1) in accordance with international guiding principles for biomedical research involving animals.

**Statistics and reproducibility**. In each experiment, the quantitative data were analyzed with 500 nanogold particles at random by using NIH ImageJ software (version 1.53c). The particle size was determined by the maximal Feret's statistical diameter (i.e., the maximal perpendicular distance between parallel tangents touching opposite sides of the profile) (Supplementary Data 1). All data are expressed as the standard deviation ± mean. Statistical significance was assessed by Student's $t$ test. $P < 0.05$ was considered statistically significant.

**Reporting summary**. Further information on research design is available in the Nature Research Reporting Summary linked to this article.

## Data availability
Original light and electron micrographs and any other information are available upon request from the corresponding author.

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

## Acknowledgements

We thank E. Nakazawa and K. Ichikawa for scientific discussions and Y. Goto and Y. Todaka for technical assistance.

## Author contributions

A.S. conceived the entire project, designed the protocol, performed low-vacuum SEM and TEM observations, wrote the manuscript, and drew all illustrations in Figs. 1, 5, and 9. T.K. performed elemental analysis by EDX-ray spectrometry. N.T., H.I., and F.A. performed sample preparation and immunocytochemistry for synaptopodin. A.Y. and Y.A. were involved in the pathological application. A.W. and T.S. performed HR-STEM analysis. N.C. and Y.H. applied in situ hybridization.

## Competing interests
The authors declare no competing interests.
