## [Peer Review File · Communications Biology]

Reviewers' comments:

Reviewer #1 (Remarks to the Author):

The authors presented a novel immunocytochemistry method that utilizes in situ gold nanoparticle labeling. Such a method is easy to apply and universal on paraffin/cryostat sections. The method presented holds high novelty and provides a new promising route for molecular localization. The experiments and results are generally well presented. However, there are still some concerns that need to be addressed:

Major problems:

1. The mechanism of gold nanoparticle formation is worth investigating. With the mechanism unrevealed, the reproducibility and the universal applicability may not be guaranteed. There are many factors that may interfere with the gold nanoparticle formation process. Other than the temperature, which is controlled in the method, and the presence of proteins, which has also been demonstrated with BSA in the paper, other factors need to be considered as well, for example: pH, which may influence the formation of gold nanoparticle; and other reductants, which may reduce chloroauric acid and lead to undesired gold nanoparticle formation and cause false-positive results.
2. Further evidence is needed to support the claim of "second growth". Presented evidence is not sufficient enough to distinguish between second growth and slow growth. Especially given the provided results of gold nanoparticle characteristics with the addition of BSA, which appeared more like slow growth. I would suggest an examination and comparison of d-spacing and FFT image between the and the branches core of gold nanoparticles.

Minor problems:

1. In the introduction section, the introduction of gold nanoparticles is not well-related to this paper and the method. Only the most basic background information of gold nanoparticles is provided, without sufficient specific introductions on how gold nanoparticles are used in the field of immunocytochemistry. More information and transactions are needed between paragraphs 2 (lines 53-60) and 3 (lines 61-66).
2. The workflow explanation (lines 77-84), along with the caption of figure 1, is a bit unclear.
3. The font size/color of the scales in figures 2, 3, 4, 6, 9, 10 are too hard to read.

Overall, though with some limitations and further investigations needed, the presented work is novel and provides promising new developments to the field.

Reviewer #2 (Remarks to the Author):

The manuscript from Sawaguchi et al describes a novel method of in situ nanogold preparation for immunocytochemistry applications. The major claim is that adding the common nanogold precursor HAuCl₄ to tissue samples exposed to 3,3'-diaminobenzidine (DAB) induces nucleation of nanogold, which subsequently grows into easily visualized nanogold particles upon exposure to a suitable humid environment over the course of 6-12 hours. This mechanism of nanogold probe preparation is certainly of interest to the field, but I believe the authors can provide more chemical details on how the nanogold nucleation and growth proceed (see specific comments below). For the most part, the statistical analysis appears valid, with two exceptions (see comments below). I would like to see more detailed text description of the experimental procedures involved in the DAB assays, conditions for nanogold growth, etc. beyond the cartoon in Figure 1 in order to provide sufficient details to reproduce

the nanogold probe synthesis. Overall, the potential applications are sufficiently important to influence future work in immunocytochemistry to warrant publication. I have the following specific comments:

1. The introduction does not have sufficient references. For instance, some general references to immunocytochemistry should be added. There should also be more discussion and references on the current state-of-the-art with nanogold probe use, preparation, and the difficulties involved on lines 64-66- perhaps the references mentioned on lines 113-114 can be discussed in further detail in the Introduction. In particular, some discussion on conventional methods for making nanogold probes is needed.

2. I would recommend pointing out that no nanogold was used for imaging in Ref. 12 (line 74).

3. On lines 87 and 137-141 the authors state that DAB and/or its products from reaction with H₂O₂ cause nucleation of HAuCl₄. I would like to see further discussion on which chemical species cause this nucleation. Is it the DAB itself, the peroxidase enzyme, or something else? I would imagine that the electrons on the amino groups of DAB could act as reducing agents.

4. Building on the previous comment, the role of H₂O₂ in nanogold growth is not discussed at all. Because H₂O₂ is used in the assays anyway, its presence likely contributes to the observed nanogold growth. The authors should add discussion regarding the role of H₂O₂ in the observed particle growth over timescales of ~hours. In particular, the statement on lines 158-159 should be revised to take into account the contribution of H₂O₂. There is extensive literature on H₂O₂ causing autocatalytic growth of already nucleated nanogold, including growth of anisotropic particles over long timescales, e.g.:

Zayats, M.; Baron, R.; Popov, I.; Willner, I., *Nano Lett.* 2005, 5, 21– 25 DOI: 10.1021/nl048547p

McGilvray, K. L.; Granger, J.; Correia, M.; Banks, J. T.; Scaiano, J. C., *Phys. Chem. Chem. Phys.* 2011, 13, 11914– 11918 DOI: 10.1039/C1CP20308H

Liu, X.; Xu, H.; Xia, H.; Wang, D., *Langmuir* 2012, 28, 13720– 13726 DOI: 10.1021/la3027804

Tangeysh, B.; Moore Tibbetts, K.; Odhner, J. H.; Wayland, B. B.; Levis, R. J., *Nano Lett.* 2015, 15, 3377– 3382 DOI: 10.1021/acs.nanolett.5b00709

5. I have two comments on Figure 5. First, it would be helpful to identify the acronyms for panel (c) in the caption. Second, in panel (e), it looks like the indicated p value between 6 hr and 12 hr isn't correct. The large error bars should indicate little if any statistical significance between 6 and 12 hr. Was the bracket supposed to be comparing 0 and 12 hr? That I would believe gives the stated p value.

6. On p. 5, lines 104-105, the authors indicate that HAuCl₄ should help enhance the contrast of DAB and provide references 17-18. They should explain the motivation behind using HAuCl₄ as a contrast agent that refs 17-18 provide. Why should the HAuCl₄ help? Is it already known to reduce to nanogold upon reaction with DAB or is it supposed to complex with the DAB?

7. On line 189, do the authors mean that the sample is 15 years old? That should be clarified (and in the Abstract too).

8. In figure 8d, It doesn't look like the difference between the NT and PBS samples at 6 hrs is statistically significant- it looks similar to the difference between the PBS at 6 hrs and the NT at 12 hrs.

Reviewer #3 (Remarks to the Author):

The paper introduces an in situ strategy of nano gold development, under hot-humid conditions, in immunoenzymatic products on universal paraffin/cryostat sections. The method is novel, and it is not only helpful for overcoming of the shortcoming of the conventional nano gold labelling methods, but also may find wide applications in applied immunocytochemistry, bridging the gap between light and electron microscopy. The work is overall convincing. There could be a limitation that the solution concentration and treatment time may need to be carefully adjusted in real applications, but it should be easily overcome-able. The paper will surely influence thinking in the field.

The reported method is user friendly, and easy to follow by other researchers of the field.

Some suggestions are listed below:

1. P5, line 91, "Fig. 1c" may be changed to "Fig. 1b"
2. P7, Line 153, "failed in dried air, which might inhibit gold monomer liquation", please check English.
3. P7, Line 158, "but the precise mechanism remains unclear because of the limited availability of experimental models": ideally, the papers of "In-situ liquid-cell TEM study of radial flow-guided motion of octahedral Au nanoparticles and nanoparticle clusters", Nano Research, 11(9) (2018) 4697; "Direct Observation of Aggregative Nanoparticle Growth: Kinetic Modeling of the Size Distribution and Growth Rate", 14 (2013) 373; and "Electron Beam Manipulation of Nanoparticles", Nano Letters , 12 (2012) 5644 might be cited and discussed.
4. Caption of Figure 5, "TEM analyses" may be changed to "Ex situ TEM analyses".

POINT-BY-POINT RESPONSES TO ALL SPECIFIC COMMENTS

We thank for all reviewers and for their positive comments. We have revised figures and text according to their specific comments.

Reviewer #1

Major problems:

1. The mechanism of gold nanoparticle formation is worth investigating. With the mechanism unrevealed, the reproducibility and the universal applicability may not be guaranteed. There are many factors that may interfere with the gold nanoparticle formation process. Other than the temperature, which is controlled in the method, and the presence of proteins, which has also been demonstrated with BSA in the paper, other factors need to be considered as well, for example: pH, which may influence the formation of gold nanoparticle; and other reductants, which may reduce chloroauric acid and lead to undesired gold nanoparticle formation and cause false-positive results.

RESPONSE:

We appreciate this important issue raised by the reviewer. In this revision, the primary screening by filter paper model assay showed a decline in the purplish tone in proportion to increasing pH value (Fig. 7a). Correspondingly, the low-vacuum SEM (Fig. 7b) demonstrated the reduction in labelling intensity on the sections. Further TEM observations (Fig. 7c) and quantitative analysis (Fig. 7d, e) clarified the reduction in secondary growth, which could be explained by the stabilization of nanogold particles at higher pH, consistent with previous reports (Ref 39-41).

To make a space for this series of experiments, we have combined original Fig. 2 and Fig 3 into revised Fig. 2.

2. Further evidence is needed to support the claim of “second growth”. Presented evidence is not sufficient enough to distinguish between second growth and slow growth. Especially given the provided results of gold nanoparticle characteristics with the addition of BSA, which appeared more like slow growth. I would suggest an examination and comparison of d-spacing and FFT image between the branches core of gold nanoparticles.

RESPONSE:

We thank the reviewer for requesting this clarification. In this revision, high-resolution scanning transmission electron microscopy (HR-STEM) demonstrated representative bright-field and

annular dark-field images of intense nanogold labelling on the processes of podocytes (Fig. 6b). Higher magnification clearly shows the details of multiple crystalline structures in the attachment region (Fig. 6c) and lattice fringes with interplanar spacing of 0.24 nm corresponding to Au (111) planes accompanied by fast Fourier transform (FFT) patterns (Fig. 6d). Moreover, HR-STEM revealed multiple crystalline structures within the spherical particles (Fig. 8c), implicating polynuclear coalescence under suppression.

In this revision, Akiko Wakui and Takeshi Sato have been added as co-authors who performed high-resolution scanning transmission electron microscopy.

Minor problems:

1. In the introduction section, the introduction of gold nanoparticles is not well-related to this paper and the method. Only the most basic background information of gold nanoparticles is provided, without sufficient specific introductions on how gold nanoparticles are used in the field of immunocytochemistry. More information and transactions are needed between paragraphs 2 (lines 53-60) and 3 (lines 61-66).

RESPONSE:

Thank you for the advice to improve the introduction section. We have revised the introduction by adding the information of conventional nanogold labelling in paragraph 3 (lines 61-71) and paragraph 4 (lines 74-78) for all readers unfamiliar with the field.

2. The workflow explanation (lines 77-84), along with the caption of figure 1, is a bit unclear.

RESPONSE:

We thank for your suggestive comment. We have revised to make it reader-friendly by changing the sentence order and paragraph break (lines 86-108). We have also indicated "Nanogold nucleation" for 0.01% H_{Au}Cl₄ treatment and "Growth & assembly" for hot-humid air incubation in Figure 1a to highlight the crucial steps for readers.

3. The font size/color of the scales in figures 2, 3, 4, 6, 9, 10 are too hard to read.

RESPONSE:

We thank for pointing this out. The font size/colour and thickness of the scales have been improved in all figures.

Reviewer #2

1. The introduction does not have sufficient references. For instance, some general references to immunocytochemistry should be added. There should also be more discussion and references on the current state-of-the-art with nanogold probe use, preparation, and the difficulties involved on lines 64-66- perhaps the references mentioned on lines 113-114 can be discussed in further detail in the Introduction. In particular, some discussion on conventional methods for making nanogold probes is needed.

RESPONSE:

Thank you for the advice to improve the introduction section. We have revised the introduction by adding the information of conventional nanogold labelling in paragraph 3 (lines 61-71) and paragraph 4 (lines 74-78) for all readers unfamiliar with the field.

2. I would recommend pointing out that no nanogold was used for imaging in Ref. 12 (line 74).

RESPONSE:

We thank for the recommendation. The application to immunocytochemical localization remained challenging due to difficulties in achieving labelling intensity with large particles. There is a trade-off relation between large particles and labelling intensity in post-embedding nanogold probe labelling (Ref 14), but large particles (> 30 nm) are indispensable for visualization under low-vacuum SEM. We have mentioned this background in the introduction section (lines 74-78).

3. On lines 87 and 137-141 the authors state that DAB and/or its products from reaction with H_2O_2 cause nucleation of $HAuCl_4$. I would like to see further discussion on which chemical species cause this nucleation. Is it the DAB itself, the peroxidase enzyme, or something else? I would imagine that the electrons on the amino groups of DAB could act as reducing agents.

RESPONSE:

We appreciate the reviewer's interest in this key point. Intensifications of DAB deposition sites have been reported with various heavy metallic ions as well as gold chloride (Ref 19, 20, 22-25). The crucial binding ability between gold chloride and the immunochemical reaction product of DAB has been indicated by energy dispersive X-ray analysis (Ref 25). However, further analysis is needed due to the lack of detailed knowledge concerning DAB polymerization and

the chemical characteristics of the resultant deposition. In this revision, we have discussed with additional references according to your kind suggestion (lines 147-157) .

4. Building on the previous comment, the role of H₂O₂ in nanogold growth is not discussed at all. Because H₂O₂ is used in the assays anyway, its presence likely contributes to the observed nanogold growth. The authors should add discussion regarding the role of H₂O₂ in the observed particle growth over timescales of ~hours. In particular, the statement on lines 158-159 should be revised to take into account the contribution of H₂O₂. There is extensive literature on H₂O₂ causing autocatalytic growth of already nucleated nanogold, including growth of anisotropic particles over long timescales, e.g.:

Zayats, M.; Baron, R.; Popov, I.; Willner, I., Nano Lett. 2005, 5, 21– 25

DOI: 10.1021/nl048547p

McGilvray, K. L.; Granger, J.; Correia, M.; Banks, J. T.; Scaiano, J. C., Phys. Chem. Chem. Phys. 2011, 13, 11914– 11918

DOI: 10.1039/C1CP20308H

Liu, X.; Xu, H.; Xia, H.; Wang, D., Langmuir 2012, 28, 13720– 13726

DOI: 10.1021/la3027804

Tangeysh, B.; Moore Tibbetts, K.; Odhner, J. H.; Wayland, B. B.; Levis, R. J., Nano Lett. 2015, 15, 3377– 3382

DOI: 10.1021/acs.nanolett.5b00709

RESPONSE:

We appreciate this important issue raised by the reviewer. In this protocol, the cell/tissue sections were treated with H₂O₂ as usual in enzyme-based immunocytochemistry. It could be speculated that residual H₂O₂ readily reduced HAuCl₄ and accelerated secondary growth (Ref 35-38). However, this speculation was contradicted by the nucleation and secondary growth observed in H₂O₂-free blotting spots. In this revision, we have discussed with additional references according to your kind suggestion (lines 191-195).

5. I have two comments on Figure 5. First, it would be helpful to identify the acronyms for panel (c) in the caption. Second, in panel (e), it looks like the indicated p value between 6 hr and 12 hr isn't correct. The large error bars should indicate little if any statistical significance between 6 and 12 hr. Was the bracket supposed to be comparing 0 and 12 hr? That I would believe gives the stated p value.

RESPONSE:

We thank the reviewer for these remarks. For the first point, the acronyms in panel (c) have been identified in the caption. Second, the statistical significance in panel (e) is correct by comparing the size of 500 nanogold particles in each group. We have revised the index from " $P < 0.001$ " to actual number " $P = 2.2 \times 10^{-12}$ " and indicated ' $N = 500$ particles/group' for its clarification. On the other hand, the index of obvious P value significance has been omitted between 0 hr and 6 hrs for clarity. The distributions of 500 particle size are shown with the actual number of Mean \pm SD, Minimum, and Maximum size in accordance with the Statistical Guidelines for Authors.

6. On p. 5, lines 104-105, the authors indicate that H₂AuCl₄ should help enhance the contrast of DAB and provide references 17-18. They should explain the motivation behind using H₂AuCl₄ as a contrast agent that refs 17-18 provide. Why should the H₂AuCl₄ help? Is it already known to reduce to nanogold upon reaction with DAB or is it supposed to complex with the DAB?

RESPONSE:

We thank the reviewer for requesting this clarification. At the beginning of this study, we initially attempted to "enhance" the contrast of DAB deposition sites with 0.1% H₂AuCl₄ solution to convert the colour signal into an electron-dense compound for electron microscopy (Ref 19, 20). Intensifications of DAB deposition sites have been reported with various heavy metallic ions as well as gold chloride (Ref 19, 20, 22-25). It has been reported that exposure to gold chloride intensifies the electron density of DAB deposition sites (Ref 20). Moreover, the crucial binding ability between gold chloride and the immunochemical reaction product of DAB has been indicated by energy dispersive X-ray analysis (Ref 25). We have revised to explain the initial motivation behind using H₂AuCl₄ (line 112-116, and line 152-155).

7. On line 189, do the authors mean that the sample is 15 years old? That should be clarified (and in the Abstract too).

RESPONSE:

We thank for the kind advice. We have clarified the meaning of "after a long lapse of 15 years" by adding a parenthesis - (i.e., 15 years old-DAB depositions) - after the phrase used in 'Results and Discussion' 'Figure legends' and 'Abstract' too, as recommended.

8. *In figure 8d, it doesn't look like the difference between the NT and PBS samples at 6 hrs is statistically significant- it looks similar to the difference between the PBS at 6 hrs and the NT at 12 hrs.*

RESPONSE:

We thank the reviewer for this concern. The statistical significance is correct between the NT and PBS samples at 6 hrs by comparing the size of 500 nanogold particles in each group. The index has been changed from asterisk mark to actual number " $P = 1.9 \times 10^{-8}$ ". The data number ' $N = 500$ particles/group' has also been indicated for clarification.

(The acronyms, NT and N/A, have been identified in the caption)

Reviewer #3

1. *P5, line 91, "Fig. 1c" may be changed to "Fig. 1b"*

RESPONSE:

We thank the reviewer for pointing out careless mistake. The typo has been corrected to "Fig. 1b".

2. *P7, Line 153, "failed in dried air, which might inhibit gold monomer liquation", please check English.*

RESPONSE:

We thank the reviewer for requesting this clarification. The description has been revised as "In fact, the secondary growth failed in incubation in dry air, which might lack the dissolved gold monomers, and in waterdrop incubation, which might dilute the dissolved gold monomers (Fig. 5b)" with a help of English editing service.

3. *P7, Line 158, "but the precise mechanism remains unclear because of the limited availability of experimental models": ideally, the papers of "In-situ liquid-cell TEM study of radial flow-guided motion of octahedral Au nanoparticles and nanoparticle clusters", Nano Research, 11(9) (2018) 4697; "Direct Observation of Aggregative Nanoparticle Growth: Kinetic Modeling of the Size Distribution and Growth Rate", 14 (2013) 373; and "Electron Beam Manipulation of Nanoparticles", Nano Letters, 12 (2012) 5644 might be cited and discussed.*

RESPONSE:

We thank for the kind advice. We have made the revision and added the references in the revised manuscript (line 180-185). It will be of great interest to apply such forefront technologies to elucidate the precise mechanism of secondary growth in hot-humid air conditions, reaching large-sized intense labelling in a short time.

4. *Caption of Figure 5, "TEM analyses" may be changed to "Ex situ TEM analyses".*

RESPONSE:

We thank the reviewer for this concern. The present TEM analyses revealed *in situ* ultrastructure of nanogold particles developed in the kidney sections. We therefore keep the original caption to avoid reader's confusion.

REVIEWERS' COMMENTS:

Reviewer #1 (Remarks to the Author):

In this revised manuscript the authors adequately addressed the concerns previously raised. The need for further research on the gold nanoparticle formation mechanism should be stressed in order to guarantee reproducibility and universal applicability. The adequate humidity range for gold nanoparticle second growth should be measured and presented. Also, exposition should be double-checked and improved for better and clearer presentation. Especially some minor problems should be adjusted. For example, in Figure 1 legend a, I believe the purpose for incubating the slide in xylene is not to remove the coverslip.

Reviewer #2 (Remarks to the Author):

The authors have addressed all of my comments and concerns in the revised version. In particular, the added discussion of the role of pH in controlling the ultimate size of nano gold particles is valuable.

I have one minor comment with regards to the added discussion about the mechanism of nano gold nucleation discussed on p. 7 and Figure 4. The added references supporting the strong binding of metal complexes to DAB are certainly helpful. My one concern is with the sentence beginning "However, further analysis is needed...". The authors should make clear that the exact mechanism of how DAB induces gold nucleation is unknown. Moreover, the invocation of "DAB polymerization" without context or references is confusing. Is it known that DAB polymerization can occur in the presence of metal complexes? If so, why might that have a role in nano gold nucleation?

POINT-BY-POINT RESPONSES TO ALL SPECIFIC COMMENTS

We thank for all reviewers and for their supportive comments. We have revised text and figures according to their comments.

Reviewer #1

In this revised manuscript the authors adequately addressed the concerns previously raised. The need for further research on the gold nanoparticle formation mechanism should be stressed in order to guarantee reproducibility and universal applicability.

RESPONSE:

We thank for your suggestive comment. Further revision has been made in the above points by adding the immunohistochemical consensus about DAB deposition, and a noteworthy report in electrochemistry (as Ref. 26) to make clear that the exact mechanism of how DAB induces gold nucleation is still open to question.

Ref 26.

Nateghi, M.R., Mosslemin, M.H. & Hadjimohammadi, H. Electrochemical preparation and characterization of poly (3,3'-diaminobenzidine): A functionalized polymer. *React. Funct.polym.* **64**, 103-109 (2005).

<Line 152-161>

In immunohistochemical staining, DAB is known to be oxidized by hydrogen peroxide (H₂O₂) in the presence of HRP that forms a brown deposition representing the location of the HRP for light microscopy. Intensifications of DAB deposition sites have been reported with various heavy metallic ions as well as gold chloride^{19,20,22-25}. The crucial binding ability between gold chloride and the immunochemical reaction product of DAB has been indicated by energy dispersive X-ray analysis²⁵. Interestingly, oxidative polymerization of DAB on gold electrode has been reported in an electrochemical study for preparation of polymeric film coated electrode²⁶. However, further analysis is needed due to the lack of detailed knowledge concerning DAB polymerization and the chemical characteristics of the resultant deposition in immunohistochemistry.

The adequate humidity range for gold nanoparticle second growth should be measured and presented.

RESPONSE:

We thank the reviewer for requesting this clarification. In this revision, the adequate humidity (= 94 - 96%) has been indicated in the text (Results and Discussion, Methods) and also in Fig. 1a and Fig. 5b.

Also, exposition should be double-checked and improved for better and clearer presentation. Especially some minor problems should be adjusted. For example, in Figure 1 legend a, I believe the purpose for incubating the slide in xylene is not to remove the coverslip.

RESPONSE:

We thank for your suggestive comment. The microscope slides were incubated in xylene for 18–48 hours to remove the coverslips by dissolving the fixed mounting medium. We have mentioned this purpose in the Methods.

Reviewer #2

The authors have addressed all of my comments and concerns in the revised version. In particular, the added discussion of the role of pH in controlling the ultimate size of nano gold particles is valuable.

I have one minor comment with regards to the added discussion about the mechanism of nano gold nucleation discussed on p. 7 and Figure 4. The added references supporting the strong binding of metal complexes to DAB are certainly helpful. My one concern is with the sentence beginning "However, further analysis is needed...". The authors should make clear that the exact mechanism of how DAB induces gold nucleation is unknown. Moreover, the invocation of "DAB polymerization" without context or references is confusing. Is it known that DAB polymerization can occur in the presence of metal complexes? If so, why might that have a role in nano gold nucleation?

RESPONSE:

We thank for your suggestive comment. Further revision has been made in the above points by adding the immunohistochemical consensus about DAB deposition, and a noteworthy report in electrochemistry (as Ref. 26) to make clear that the exact mechanism of how DAB induces gold nucleation is still open to question.

Ref 26.

Nateghi, M.R., Mosslemin, M.H. & Hadjimohammadi, H. Electrochemical preparation and characterization of poly (3,3'-diaminobenzidine): A functionalized polymer. *React. Funct.polym.* **64**, 103-109 (2005).

<Line 152-161>

In immunohistochemical staining, DAB is known to be oxidized by hydrogen peroxide (H_2O_2) in the presence of HRP that forms a brown deposition representing the location of the HRP for light microscopy. Intensifications of DAB deposition sites have been reported with various heavy metallic ions as well as gold chloride^{19,20,22-25}. The crucial binding ability between gold chloride and the immunochemical reaction product of DAB has been indicated by energy dispersive X-ray analysis²⁵. Interestingly, oxidative polymerization of DAB on gold electrode has been reported in an electrochemical study for preparation of polymeric film coated electrode²⁶. However, further analysis is needed due to the lack of detailed knowledge concerning DAB polymerization and the chemical characteristics of the resultant deposition in immunohistochemistry.